# Atherogenic dyslipidemia and associated risk factors among hypertensive patients of five health facilities in Northeast Ethiopia

**Ousman Mohammed** *, **Ermiyas Alemayehu, Endris Ebrahim, Mesfin Fiseha, Alemu Gedefie, Abdurrahman Ali, Hussen Ebrahim, Mihret Tilahun**

Department of Medical Laboratory Sciences, College of Medicine and Health Sciences, Wollo University, Dessie, Ethiopia

* ousmanabum@gmail.com

## Abstract

### Background

One of the major risk factors for cardiovascular disease is atherogenic dyslipidemia. There was, however, little information available in Ethiopia. Therefore, the purpose of this study was to estimate the prevalence of atherogenic dyslipidemia and related risk factors in Northeast Ethiopian hypertension patients.

### Materials and methods

A systematic random sampling technique was used to perform a cross-sectional study at an institution with 384 chosen participants. A structured questionnaire was used to collect the socio-demographic, anthropometric, lifestyle, and clinical characteristics of the respondents. Student's t-test, Mann-Whitney test, and Pearson's Chi-square test were employed to compare groups based on the type of data. Furthermore, Bivariate and multivariable logistic regression analyses were performed to identify factors independently associated with dyslipidemia. Crude and adjusted odds ratios and their corresponding 95% Confidence Intervals (CI) were computed. In all cases, statistical significance was declared at p <0.05.

### Results

The majority (93.2%; 95%CI: 90.6–95.6) of patients had at least one atherogenic dyslipidemia. The prevalence of elevated total cholesterol (TC), elevated triglyceride (TG), raised low-density lipoprotein cholesterol (LDL-c), and reduced high-density lipoprotein cholesterol (HDL-c) were 47.7%, 50.3%, 44.3%, and 59.6%, respectively. Being ≥ 40 years were at higher risk for having elevated levels of TC (AOR: 3.22, 95% CI: 2.40–4.32), TG (AOR: 2.30, 95% CI: 1.61–3.79), and LDL-c (AOR: 4.68, 95% CI: 2.0–10.95) than those who were below 40 years. Obese participants were more likely to have high concentrations of TC (AOR: 2.57, 95%CI: 2.10–3.22), LDL-c (AOR: 3.13, 95% CI: 1.97–5.10), HDL-c (AOR: 2.71, 95% CI: 1.77–4.58), and TG (AOR: 2.23, 95%CI: 1.79–4.16).

**Data Availability Statement:** All relevant data are within the paper and its Supporting information files.

**Funding:** The authors received no specific funding for this work.

**Competing interests:** The authors declare that they have no potential competing interests.

## Conclusion

This study revealed that a high prevalence of atherogenic dyslipidemia. Thus, to prevent atherogenic dyslipidemia, it is crucial to create routine blood lipid testing programs and carry out suitable intervention programs focused on risk factor reduction.

## Introduction

Globally, it is estimated that one billion adults live with hypertension; this figure is predicted to be more than 1.5 billion by the year 2025. Furthermore, at least 45% of fatalities from heart disease and 51% of deaths from stroke were caused by hypertension [1]. Due to inadequate levels of therapy and management of hypertension in Africa, the high burden of hypertension is the main cause of mortality and morbidity linked with cardiovascular disorders globally [2–5]. In Ethiopia, hypertension is one of the most common public health burdens. The pooled prevalence of hypertension in the Ethiopian population was calculated to be 19.6 percent [6].

More than half of the 17.4 million fatalities per year that occur worldwide are caused by cardiovascular illnesses, which are connected to hypertension. The research suggests that responsible authorities should pay more attention to the cardiovascular health of African populations living in both rural and urban areas [7, 8]. According to the etiology and prognosis of atherosclerosis and cardiovascular disease (CVD), hypertension and atherogenic dyslipidemia frequently coexist [9–12]. One of the primary risk factors for the development of atherosclerosis is atherogenic dyslipidemia. The prevalence of cardiovascular disorders has increased as a result of the expansion of atherosclerosis. Thus, it is clear that the frequent co-occurrence of hypertension, atherosclerotic dyslipidemia, and other metabolic abnormalities in patients increases the risk of heart failure and CVD-related morbidity and death [13–16].

Currently, CVD is one of the most common causes of death worldwide and is on the rise [17]. Compared to people with normal lipid levels, those with atherogenic dyslipidemia are more likely to develop CVD [18]. High levels of total cholesterol (TC), hypertriglyceridemia, low-density lipoprotein cholesterol (LDL-c), lower HDL-c, and an increased atherogenic index TC/HDL-c ratio are all indicators of atherogenic dyslipidemia [19–22]. Atherosclerosis, which is known as the main risk factor for stroke, peripheral vascular disease, and coronary heart disease (CHD), is made more likely by elevated blood levels of certain lipids [23]. LDL-c may have an impact on the onset of atherosclerosis [22]. On the other hand, HDL-c plays a role in the reverse transport of cholesterol, which decreases the risk of atherosclerotic CVD [22, 24].

Patients with coexisting cardiovascular risk factors, such as hypertension, have a substantially greater prevalence of dyslipidemia [25, 26]. A number of risk factors were linked to dyslipidemia among hypertension patients, according to empirical data from earlier literature. These risk variables included sedentary behavior, age, gender, obesity, smoking, diabetes, and a poor diet of fruits and vegetables [27, 28]. Although non-healthy food and people's behaviors are linked to roughly 80% of dyslipidemia [29], this percentage is still high. In addition, Kifle Z et al. and Hirigo A et al., respectively, observed high rates of dyslipidemia among Ethiopian hypertensive patients (48.4% and 90.8%) [27, 28].

Evidence suggested that the rising prevalence of atherogenic dyslipidemia and hypertension in patients may significantly deteriorate their state of health [10, 12, 30, 31]. In the majority of developing nations, including Ethiopia, CVD-related diseases and hypertension have recently become major public health issues [3, 4, 27]. There were few data on the prevalence of

dyslipidemia and related variables in hypertension patients in Ethiopia, despite its high prevalence and related complication in this population. To our knowledge, we found only a few published studies that were conducted in one healthcare setting. As a result, they might not be representative of the general public. The current study's objective was to evaluate atherogenic dyslipidemia and the factors that contribute to it among hypertensive patients at five healthcare facilities in northeast Ethiopia.

## Materials and methods

### Study design, setting and period

The current institution-based cross-sectional study was conducted at the selected hospitals in Dessie town from June to October 2021. The town has a cold temperature and is located on a mountain crest. Its entire area is 15.08 km2, and its distance from Addis Ababa, the county headquarters, and Bahir Dar, the capital of the Amhara regional state, respectively, is 401 km and 471 km. Eight health facilities, three private general hospitals, five higher private clinics, one general hospital, one comprehensive specialized hospital, and one general hospital are all located in Dessie town and provide medical services to the surrounding areas [32]. For the purpose of this study five (namely, Dessie comprehensive specialized hospital, Borumeda general hospital, Selam general hospital, Ethio general hospitals, and Bati general hospitals) health facilities that provide chronic diseases services were selected.

### Study population and legibility criteria

Patients with hypertension who visited the chronic department of particular hospitals during the study period were included. Based on a single population proportion calculation and a 48.4% overall prevalence of dyslipidemia in hypertension patients, the sample size was determined [27]. With a 95% confidence level, the expected margin of error (d) was calculated at a level of 0.05. Thus, the calculated sample size was 384. The overall sample size was appropriately distributed based on the number of registered hypertension patients in each healthcare setting because the data were gathered from five distinct healthcare facilities. A methodical random sampling strategy was used to choose the respondents. All volunteer hypertensive patients age ≥18years-old who had a regular follow-up were eligible in the 1study. However, physiological and pathological factors that could alter serum lipid profiles, such as pregnancy, taking lipid-altering drugs, and antihyperlipidemic medications, were excluded because the study was designed to assess dyslipidemia. Moreover, critically ill and those with mental problems and unable to communicate were excluded from the study.

### Data collection and quality control

Data were collected through a structured, validated, and pretested face-to-face interviewer-administered questionnaire. The questionnaire contained information on socio-demographics, history of co-morbidity, the habit of physical exercise, behavioral habits (alcohol consumption and smoking), etc. Two well-trained data collectors (BSc nurses) were recruited for each health facility. They were given training for two days on the method of extracting the needed information, how to fill the information on a structured questionnaire, and the ethical aspect of approaching the participants as well as the aim of the study and the contents of the instruments. The data collection material was pre-tested on 10% of the sample size to check completeness, consistency, and applicability and was modified accordingly. The researchers were making spot-checks of at least 5 questionnaires per day. Reviewing the completed questionnaire by the data collectors ensures completeness and consistency of the information that was

collected. Quality control was carried out during the pre-analytical, analytical stages, and post-analytical phases. Furthermore, patient samples were evaluated alongside analytical stage quality control checks, which included quality control materials (Normal, Low, and High) to detect if analytic errors had occurred.

### Anthropometric measurement and biochemical analysis

Following a standardized protocol, weight, and height were measured by trained collectors. Standing heights were taken without shoes to the closest 0.1 cm using well-situated stadiometers. The weights were measured to the closest 0.1 kg using a digital balance. Weights were measured without heavy clothes and shoes, while heights were measured without shoes to get an accurate measurement. After 5 minutes of rest, blood pressure was checked in triplicate, with subsequent readings taken 5 minutes apart. Additionally, body mass index (BMI) was calculated as the product of weight (kg) and height (meters), squared ($kg/m^2$). Participants with a BMI lower than 18.5 kg/m2 were considered as underweight; between 18.5 and 24.9 kg/m2 as normal; between 25.0 and 29.9 kg/m2 as overweight and 30.0 kg/m2 and above as obese [33]. Waist circumference (WC) was measured at the level of the iliac crest and the level of the umbilicus in cm to evaluate abdominal obesity, and raised WC was defined as ≥94 cm for men and ≥80 cm for women [34]. The average values from each set of three duplicate anthropometric measurements were utilized to conduct the analysis.

After obtaining consent, around 5 ml of venous blood sample was collected after overnight fasting for TG, HDL-c, LDL-c, and TC tests. The blood sample was clotted for 30 minutes. Then, the serum sample was separated from the Nunc tube following centrifuged for 5 minutes at 4000 revolutions per minute. Lipid profile parameters were analyzed with a fully automated clinical chemistry analyzer using the direct endpoint enzymatic process. All samples were analyzed within 24 hours with the same analyzer to minimize assay variation. Before sample analysis, the machine was checked using controls and blank on a daily basis. The cut-offs for abnormal serum lipid levels were: ≥200 mg/dL for total cholesterol (TC), for triglyceride (TG) concentrations of ≥150 mg/dL, for (LDL-c)>130 mg/dL, and HDL-c <40 mg/dL based on the National Cholesterol Education Program (NCEP) reference limits. According to NCEP the applied the cut-off value for TC/HDL-c ratio was ≥5. Hence, according to the NCEP guidelines, individuals should have at least one of the lipid parameters that become abnormal to be categorized under the presence of dyslipidemia [22].

### Data analysis

The statistical analysis was performed using the SPSS version 23 statistics package for social sciences. Data were summarized as means/median ± standard deviation and proportion (percentages) for continuous and qualitative data, respectively. Comparisons between groups were done using Student's t-test and $X^2$, respectively, for continuous and categorical data respectively. Besides, the normality of the continuous variables was checked and the Mann-Whitney test was used for skewed distribution. Furthermore, bivariate and multivariable logistic regression analyses were performed to identify factors independently associated with dyslipidemia. Crude and adjusted odds ratios and their corresponding 95% Confidence Intervals (CI) were computed. In all cases, statistical significance was declared at p <0.05.

### Ethics approval and consent to participate

The study was approved by the Institutional Review Board of Wollo University, College of Medicine and Health Sciences, and ethical clearance was obtained. The appropriate health facilities received official letters of collaboration, and consent was acquired. Prior to collecting

any data, each study participant gave their written informed consent after being fully aware of the study's protocols and the involvement was completely voluntary.

## Results

### Prevalence of dyslipidemia according to socio-demographic characteristics of study participant's

A total of 384 respondents were included in the current study with a 100% response rate. Among them, 202 (52.6%) were men. More than two-thirds (67.2%) of the hypertensive individuals were 40 years and above, while 32.8% were in the age range of 18–39 years with a mean age of 46.5±12.7 years. Similarly, more than half 213(55.5%) of the hypertensive peoples were permanent residents of urban areas. Of the total hypertensive individuals, the highest of prevalence lipid abnormalities was seen for reduced HDL-C level (59.6%), followed by elevated triglycerides (50.3%), elevated total cholesterol (47.7%), and elevated LDL-C (44.3%) in both sexes. Likewise, the magnitude of all serum lipid profile derangements was significantly different across gender and age groups (P<0.05). A higher percentage of female hypertensive individuals had elevated lipid profiles (P<0.05). Furthermore, the urban residence participants had significantly higher serum lipid abnormalities of 122(57.3%), 113 (55.0%), 119(55.9%), and 133(62.4%) for TC, TG, LDL-c, and HDL-c respectively (P <0.05) (Table 1).

### Magnitude of dyslipidemia according to comorbidity and lifestyle practices of participants

Regarding the respondents' lifestyle, 287(74.7%) and 311(81.0%) reported that they did not regularly eat fruits/vegetables. While 139 people (36.2%) and 34 (8.9%) had experienced overweight and obesity, respectively, half of the hypertensive individuals (51.6%) had maintained normal body weight. Additionally, 319 hypertensive patients (83.1%) reported chewing tobacco, 60 (15.6%) claimed now smoking cigarettes and 67 participants (17.4%) reported currently drinking alcohol. Only 239 (62.2%) of the individuals consistently took their antihypertensive medications.

Interestingly, among those hypertensive patients who engaged in sedentary physical activity, the prevalence of atherogenic dyslipidemia was highest for TC, TG, LDL-c, and HDL-c at 54.0%, 57.9%, 52.1%, and 66.9%, respectively (P<0.05). Additionally, the prevalence of atherogenic dyslipidemia varied from 76.5% to 91.2%, among people who were overweight or obese, while it ranged from 26.3 to 51.0% in those with normal body weight (p <0.005). Furthermore, a raised waist circumference, long-term hypertension, and familial history of hypercholesterolemia were all consistently associated with low HDL-c (p <0.005). Similarly, among those who reported a familial history of hypercholesterolemia, TC, TG, LDL-c, and HDL-c lipid derangements were prevalent in 80.8%, 74.4%, 69.2%, and 84.6% of cases, respectively. Patients with DM co-morbidities had the highest prevalence of elevated LDL-c (72.4%) with a P-value<0.005. Additionally, as shown in the table below, atherogenic dyslipidemia varied from 66.0% for LDL-c to 76.7% for HDL-cholesterol (P<0.05) among people with hypertension for at least 10 years (Table 2).

Female hypertensive patients had significantly higher mean serum concentrations of TC, LDL-c, HDL-c, and TG (p-value<0.05). Besides, the derived mean ±standard deviation (SD) of the TC, LDL-c, and HDL-c and the median value of TG were 198.6±54.2, 123.3±41.2, 40.2 ±10.4, 170±109.7 for combined sexes respectively (Table 3).

**Table 1. Distribution of dyslipidemia by socio-demographic characteristics of hypertensive patients in Northeast Ethiopia, 2021 (n = 384).**

| Variables | | Frequency (%) | TC ≥ 200 mg/dL N (%) | TG ≥ 150 mg/dL N (%) | LDL-c> 130 mg/dL N (%) | HDL-c < 40 mg/dL N (%) |
|---|---|---|---|---|---|---|
| Age in year | 18–39 | 126(32.8) | 18(14.3) | 51(40.5) | 21(16.7) | 64(50.8) |
| | ≥40 | 258(67.2) | 165(64.0) | 142(55.0) | 149(57.8) | 165(64.0) |
| | P-value | **<0.001** | **0.001** | **<0.001** | **0.02** | |
| Sex | Female | 182(47.4) | 92(50.5) | 108(59.3) | 87(47.8) | 119(65.4) |
| | Male | 202(52.6) | 91(45.1) | 85(42.1) | 83(41.1) | 110(55.4) |
| | Combined | 384 (100) | 183 (47.7) | 193 (50.3) | 170 (44.3) | 229 (59.6) |
| | P-value | **0.02** | **0.03** | **0.01** | **0.03** | |
| Residence | Urban | 213(55.5) | 122(57.3) | 113(55.0) | 119(55.9) | 133(62.4) |
| | Rural | 171(44.5) | 81(47.4) | 80(46.8) | 78(45.6) | 96(56.1) |
| | P-value | **0.01** | **0.004** | **0.03** | **0.02** | |
| Marital status | Single | 102(26.6) | 32(31.4) | 45(44.1) | 33(32.4) | 53(52.0) |
| | Married | 212(55.2) | 111(52.4) | 109(51.4) | 99(46.7) | 132(62.3) |
| | Divorced | 45(11.7) | 26(57.8) | 22(48.9) | 15(33.3) | 26(57.8) |
| | Widowed | 25(6.5) | 14(56.0) | 17(68.0) | 13(52.0) | 18(72.0) |
| | P-value | 0.52 | 0.87 | **0.03** | 0.19 | |
| Occupations | Government | 114(29.7) | 45(39.5) | 54(47.4) | 37(32.4) | 67(58.8) |
| | Nongovernment | 63(16.4) | 28(44.4) | 36(57.1) | 25(39.7) | 34(54.0) |
| | Self employed | 105(27.3) | 53(50.5) | 55(52.4) | 51(48.6) | 71(67.6) |
| | Student | 11(2.9) | 2(18.2) | 4(36.4) | 3(27.3) | 6(54.5) |
| | Farmer | 91(23.7) | 55(60.4) | 44(48.4) | 54(59.3) | 51(56.0) |
| | P-value | **0.009** | 0.23 | 0.05 | 0.21 | |
| Monthly income in ETB | >3000 | 122(31.8) | 56(45.9) | 58(47.5) | 54(44.3) | 70(57.4) |
| | 2001–3000 | 99(25.8) | 47(47.5) | 52(52.5) | 43(43.4) | 62(50.8) |
| | 1000–2000 | 86(22.4) | 44(51.2) | 47(54.6) | 33(38.4) | 51(59.3) |
| | <1000 | 77(20.1) | 36(46.8) | 36(46.8) | 40(51.9) | 46(59.7) |
| | P-value | 0.89 | 0.14 | 0.53 | 0.89 | |
| Educational status | Secondary and above | 67(17.4) | 28(41.8) | 31(46.3) | 23(34.3) | 33(49.2) |
| | Primary | 132(34.4) | 57(43.2) | 66(50.0) | 48(36.4) | 80(60.6) |
| | Illiterate | 185(48.2) | 96(52.0) | 90(48.6) | 93(50.3) | 112(60.5) |
| | P-value | 0.27 | 0.84 | 0.21 | 0.72 | |

Note: TC: Total cholesterol, TG: Triglycerides, LDL-c: Low-Density Lipoprotein Cholesterol, HDL-c: High-Density Lipoprotein Cholesterol: P-value determined using Chi-square test

## Co-occurrence of the lipid abnormalities and raised TC/HDL-c ratio

In the study subjects, the combined elevation of TC+TG was 68 (37.4%), 50 (24.8%) for males, and 50 (24.8%) for females. Similarly, the overall prevalence of elevated TC+LDL-c was 76 (37.6 percent) for males and 73 (40.1%) for female hypertensive patients. Besides, the prevalence of three lipid profile derangements (TC+TG+LDL-c) in a single individual was 89 (23.2%), while 67 (17.4%) of the hypertensive subjects had exhibited abnormalities in all four serum lipids. The present study revealed that the majority (93.2%; 95%CI: 90.6–95.6) of the hypertensive patients had experienced dyslipidemia in at least one lipid profile that is compatible with the diagnosis of dyslipidemia. Moreover, this work noted that more than half (52.3%) of the hypertensive patients had a raised TC/HDL-c ratio (Table 4).

**Table 2. Distribution of dyslipidemia by behavioural and lifestyle characteristics and other medical profiles of hypertensive patients in Northeast Ethiopia, 2021 (n = 384).**

| Variables | | Frequency (%) | TC ≥ 200 mg/dL N (%) | TG ≥ 150 mg/dL N (%) | LDL-c > 130 mg/dL N (%) | HDL-c < 40 mg/dL N (%) |
|---|---|---|---|---|---|---|
| Regular physical activity | Yes | 73(19.0) | 15(20.5) | 13(17.8) | 8(10.9) | 21(28.8) |
| | No | 311(81.0) | 168(54.0) | 180(57.9) | 162(52.1) | 208(66.9) |
| | P-value | **0.01** | **<0.001** | **0.003** | **0.04** |
| Eating habits of fruits/vegetables | Yes | 97(25.3) | 30(30.9) | 29(29.9) | 16(16.5) | 24(24.7) |
| | No | 287(74.7) | 153(53.3) | 164(57.1) | 154(53.6) | 205(71.4) |
| | P-value | **0.007** | **0.03** | **0.02** | 0.06 |
| Body mass index | Underweight | 13(3.4) | 5(38.5) | 7(53.8) | 4(30.8) | 5(38.5) |
| | Normal | 198(51.6) | 55(27.8) | 66(33.3) | 52(26.3) | 101(51.0) |
| | Overweight | 139(36.2) | 93(66.9) | 89(64.0) | 86(61.8) | 102(73.4) |
| | Obesity | 34(8.9) | 30(88.2) | 31(91.2) | 28(82.4) | 26(76.5) |
| | P-value | **0.001** | **<0.001** | **0.004** | **0.003** |
| Waist circumference (cm) (Men/women) | < 94/80 | 243(63.3) | 100(41.2) | 109(44.8) | 95(39.1) | 139(57.2) |
| | ≥94/80 | 141(36.7) | 103(73.0) | 94(66.7) | 105(74.5) | 110(78.0) |
| | P-value | **0.003** | **<0.001** | **0.01** | **0.002** |
| Current alcohol consumption | No | 317(82.6) | 147(46.4) | 156(49.2) | 136(42.9) | 190(60.0) |
| | Yes | 67(17.4) | 36(53.7) | 37(55.2) | 34(50.7) | 39(58.2) |
| | P-value | 0.07 | 0.36 | 0.29 | 0.29 |
| Current cigarettes smoking | No | 324(84.4) | 157(48.4) | 150(46.3) | 131(40.4) | 188(58.0) |
| | Yes | 60(15.6) | 26(43.3) | 43(71.7) | 39(65.0) | 41(68.3) |
| | P-value | **0.004** | **0.04** | **0.02** | 0.07 |
| Habit of chat chewing | No | 65(16.9) | 33(50.8) | 34(52.3) | 15(23.1) | 40(61.5) |
| | Yes | 319(83.1) | 169(53.0) | 159(49.8) | 155(48.6) | 189(59.2) |
| | P-value | 0.06 | 0.54 | 0.34 | 0.07 |
| Adherence to antihypertensive medicines | Yes | 239(62.2) | 120(50.4) | 80.8(33.8) | 59(24.6) | 84(35.2) |
| | No | 245(63.8) | 163(66.5) | 146(59.6) | 136(55.5) | 180(73.5) |
| | P-value | **0.03** | **0.009** | **0.03** | **0.04** |
| Duration of hypertension in year | <5 | 157(40.9) | 32(20.4) | 45(28.7) | 22(14.0) | 60(38.2) |
| | 5–9 | 77(20.0) | 43(55.8) | 55(71.4) | 49(63.6) | 54(70.1) |
| | ≥10 | 150(39.1) | 108(72.0) | 113(75.3) | 99(66.0) | 115(76.7) |
| | P-value | **<0.001** | **0.01** | **0.01** | **0.007** |
| Comorbidity with hypertension | No diseases | 294(76.5) | 143(48.6) | 150(51.0) | 117(39.8) | 175(59.5) |
| | Renal | 9(2.3) | 5(55.5) | 6(66.7) | 6(66.7) | 7(77.8) |
| | Diabetes | 76(19.8) | 46(60.5) | 47(61.8) | 55(72.4) | 47(61.8) |
| | Liver | 6(1.4) | 3(50.0) | 3(50.0) | 2(33.3) | 2(33.3) |
| | P-value | **0.001** | **0.005** | **0.001** | **0.02** |
| Family history of hypercholestronimia | No | 306(79.7) | 120(39.2) | 135(44.1) | 116(37.9) | 183(59.8) |
| | Yes | 78(20.3) | 63(80.8) | 58(74.4) | 54(69.2) | 66(84.6) |
| | P-value | **0.04** | **0.03** | **0.02** | **0.004** |

Note: TC: Total cholesterol, TG: Triglycerides, LDL-c: Low-Density Lipoprotein Cholesterol, HDL-c: High-Density Lipoprotein Cholesterol: P-value determined using Chi-square test

## Predictors of dyslipidemia among hypertensive patients

In the bivariate analysis model, many predictors such as sex, age, marital status, occupation, taking antihypertensive medicine, duration of hypertension, family history of

**Table 3. The mean/median values of serum lipid profile and other risk factors stratified by gender (n = 384).**

| Variables | All | Women | Men | p-value |
|---|---|---|---|---|
| Age | 46.5±12.7 | 45.5±13.0 | 47.6±12.3 | 0.1 |
| TC | 198.6±54.2 | 201.0±44.1 | 196.6±33.8 | **0.02** |
| TG | 170±109.7 | 170±89.3 | 135±79.7 | **0.04[¶]** |
| LDL-c | 123.3±41.2 | 125.7±23.1 | 121.1±32.8 | **0.01** |
| HDL-c | 40.2±10.4 | 41.1±12.1 | 39.1±11.4 | **0.01** |
| BMI | 24.4±3.8 | 24.2±4.0 | 24.6±3.6 | **0.02** |
| WC (cm) | 93.1±11.2 | 83.4±12.8 | 84.2±9.0 | **<0.001[¶]** |
| SBP/DBP(mmHg) | 146.7±17.1/86.3±13.5 | 147±13.8/87.2±9.7 | 143.8±10.5/88.4±11.7 | 0.44/0.06 |

Note: TC: Total cholesterol, TG: Triglycerides, LDL-C: Low-Density Lipoprotein Cholesterol HDL-C: High-Density Lipoprotein Cholesterol, BMI: Body mass index, WC: Waist Circumference, DBP: diastolic blood pressure, SBP: Systolic Blood Pressure; P-value determined using Student's t-test,

[¶]P-value determined using Mann-Whitney Test

hypercholesterolemia, BMI, current cigarette smoking, and comorbidity were recruited as risk factors for most of the lipid profile derangements (P-value<0.05). Participants whose age ≥ 40 years were at higher risk for having elevated levels of TC and TG with a value of (AOR: 3.22, 95% CI: 2.40–4.32, P-value<0.001), and (AOR: 2.30, 95% CI: 1.21–3.79, P-value = 0.04) than those who were below 40 years of age respectively. Regarding sex, female hypertensive individuals are at higher risk for having elevated concentrations of TC (AOR: 2.02, 95% CI: 1.26–3.70, P-value = 0.02), and TG (AOR: 1.52, 95% CI: 1.02–1.92, P-value = 0.005) than male counterparts. Moreover, participants who had sedentary lifestyles are at more risk for having elevated atherogenic TC and TG levels (AOR: 2.01, 95% CI: 1.52–2.89, P-value = 0.03), and TG (AOR: 1.94, 95% CI: 1.25–2.69, P-value = 0.04), respectively. Additionally, obese people are more likely to have high concentrations of TC (AOR: 2.57, 95%CI: 1.97–3.22, P-value = 0.01) and TG (AOR: 2.23, 95%CI: 1.29–4.16, P-value = 0.03). Likewise, multivariate analysis revealed that current cigarette smoking and the habit of not eating fruits and vegetables were significantly associated with elevated TC levels but not with elevated TG levels (Table 5).

The odds of aberrant LDL-c and HDL-c were also 4.68 (AOR: 4.68, 95%CI: 2.0–10.95) and 1.22 (AOR: 1.22, 95%CI: 0.58–2.56) times higher among patients aged 40 years and older, respectively, compared to subjects aged below 40 years, in the multivariable logistic regression model. Participants with a history of current smoking had 1.75 (AOR: 1.75, 95%CI: 1.19–2.43)

**Table 4. Co-occurrence of the four lipid derangements among hypertensive patients stratified by gender in Northeast Ethiopia, 2021.**

| Combined lipid derangements | Female (n = 182) | | Male (n = 202) | | Combined sexes (n = 384) | |
|---|---|---|---|---|---|---|
| | Yes N (%) | No N (%) | Yes N (%) | No N (%) | Yes N (%) | No N (%) |
| TC+TG elevated | 68 (37.4) | 114 (62.6) | 50 (24.8) | 152 (75.2) | 118 (30.7) | 266 (69.3) |
| TC+LDL-c elevated | 73 (40.1) | 109 (59.9) | 76 (37.6) | 126 (62.4) | 149 (38.8) | 235 (61.2) |
| Elevated TC +reduced HDL-c | 67 (36.8) | 115 (63.2) | 62 (30.7) | 140 (69.3) | 129 (33.6) | 255 (66.4) |
| TC+TG+LDL-c elevated | 49 (26.9) | 133 (73.1) | 40 (19.8) | 162 (80.2) | 89 (23.2) | 295 (76.8) |
| TC+TG+LDL-c+HDL-c | 37 (20.3) | 145 (79.7) | 30 (14.9) | 172 (85.1) | 67 (17.4) | 317 (82.6) |
| Overall prevalence of dyslipidemia in at least one lipid profile | 167 (91.8) | 15 (8.2) | 191 (94.6) | 11 (5.4) | 358 (93.2) | 26 (6.8) |
| TC/HDL-c ratio ≥5 | 111 (61.0) | 71 (39.0) | 90 (44.6) | 112 (55.4) | 201 (52.3) | 183 (47.7) |

Abbreviations: TC: Total cholesterol, TG: Triglycerides, LDL-c: Low-Density Lipoprotein Cholesterol, HDL-c: High-Density Lipoprotein Cholesterol

**Table 5. Multivariable logistic regression analysis of factors associated with elevated serum total cholesterol and triglycerides levels among hypertensive patients in Northeast Ethiopia, 2021 (n = 384).**

| Variables | | TC ≥ 200 mg/dL | | | | TG ≥ 150 mg/dL | | | |
|---|---|---|---|---|---|---|---|---|---|
| | | COR (95%CI) | P-value | AOR (95%CI) | p-value | COR (95%CI) | P-value | AOR (95%CI) | P-value |
| Age in year | 18–39 | 1 | | 1 | | 1 | | 1 | |
| | ≥40 | 2.09(1.16–3.70) | <0.001 | 3.22(2.240–4.32) | <0.001 | 1.57(1.06–1.89) | 0.01 | 2.30(1.21–3.79) | 0.04 |
| Sex | Male | 1 | | 1 | | 1 | | 1 | |
| | Female | 1.25(0.84–1.86) | 0.03 | 2.02(1.26–3.70) | 0.02 | 2.01(1.34–3.02) | 0.001 | 1.52(1.02–1.92) | 0.005 |
| Residence | Rural | 1 | | 1 | | 1 | | 1 | |
| | Urban- | 1.92(1.02–2.53) | 0.006 | 1.68(1.03–2.36) | 0.03 | 1.28(0.86–1.92) | 0.002 | 2.06(1.56–3.72) | 0.006 |
| Marital status | Single | 1 | | 1 | | 1 | | 1 | |
| | Married | 0.36(0.15–0.89) | 0.03 | 0.66( | 0.30 | 1.38(1.15–1.96) | 0.04 | 0.97(0.57–1.66) | 0.56 |
| | Divorced | 0.86(0.38–2.00) | | 0.86(0.66–1.14) | | 1.50(1.21–2.10) | | 0.97(0.44–2.11) | 0.63 |
| | Widowed | 1.08(0.40–2.90) | | | | 1.45(1.16–2.25) | | 1.78(0.65–4.88) | 0.21 |
| Occupations | Government employee and Student | 1 | | 1 | | 1 | | 1 | |
| | Nongovernment | 1.43(0.24–0.75) | 0.01 | 1.11(0.56–2.18) | 0.51 | 1.04(0.60–1.81) | 0.04 | 1.58(0.79–3.17) | 0.07 |
| | Self employed | 1.52(0.27–1.00) | | 1.21(0.55–2.69) | | 1.22(0.72–1.08) | | 1.21(0.68–2.16) | 0.04 |
| | Farmer | 1.67(0.38–1.20) | | 1.95(1.08–3.11) | | 1.48(0.80–2.75) | | 1.75(1.40–2.42) | 0.03 |
| Monthly income in ETB/month | >3000 | 1 | | | | 1 | | | |
| | 2001–3000 | 0.97(0.54–1.71) | 0.90 | | | 1.22(0.72–2.08) | 0.66 | | |
| | 1000–2000 | 1.03(0.57–1.87) | | | | | | | |
| | <1000 | 1.20(0.64–2.21) | | | | 1.33(0.67–2.31) | | | |
| | | | | | | 0.97(0.55–1.72) | | | |
| Educational status | Secondary and above | 1 | | | | 1 | | | |
| | Primary | 0.75(0.43–1.32) | 0.27 | | | 0.81(0.25–1.46) | 0.65 | | |
| | Illiterate | 0.70(0.45–1.10) | | | | 0.77(0.24–1.35) | | | |
| Regular physical activity | Yes | 1 | | 1 | | 1 | | 1 | |
| | No | 2.15(1.57–2.98) | 0.04 | 2.01(1.25–2.89) | 0.03 | 1.70(1.42–2.16) | 0.01 | 1.94(1.52–2.69) | 0.04 |
| Eating habits of fruits/vegetables | Yes | 1 | | 1 | | 1 | | | |
| | No | 1.30(0.82–2.07) | 0.03 | 2.31(1.36–3.12) | 0.02 | 0.93(0.59–1.48) | 0.77 | | |
| Body mass index | Underweight | 1 | | 1 | | 1 | | 1 | |
| | Normal | 0.26(0.07–1.00) | <0.001 | 1.33(0.88–1.40) | 0.01 | 0.74(0.24–2.29) | 0.02 | 0.42(0.12–1.44) | 0.52 |
| | Overweight | 1.27(0.82–1.58) | | 1.70(1.07–2.93) | | 1.03(0.33–3.23) | | 1.60(0.91–2.07) | 0.04 |
| | Obesity | 1.52(0.93–2.16) | | 2.57(1.97–3.22) | | 1.96(1.17–3.47) | | 2.23(1.29–4.16) | 0.03 |
| Waist circumference (cm) (Men/women) | <94/80 | 1 | | 1 | | 1 | | 1 | |
| | ≥94/80 | 2.05(1.14–3.12) | 0.001 | 2.17(1.23–3.83) | 0.007 | 1.81(1.09–2.76) | 0.006 | 3.20(1.88–5.45) | <0.001 |
| Habit of drinking alcohol | No | 1 | | | | 1 | | | |
| | Yes | 0.74(0.14–1.26) | 0.27 | | | 1.27(0.75–2.16) | 0.37 | | |
| Current cigarettes smoking | No | 1 | | 1 | | 1 | | 1 | |
| | Yes | 1.79(1.14–2.41) | 0.04 | 1.49(1.14–2.87) | 0.04 | 1.96(1.55–2.65) | 0.03 | 1.84(1.45–2.57) | 0.09 |
| Habit of chat chewing | No | 1 | | | | 1 | | 1 | |
| | Yes | 1.16(0.68–1.98) | 0.58 | | | 1.91(1.53–2.54) | 0.04 | 1.20(0.66–2.17) | 0.56 |

*(Continued)*

**Table 5.** (Continued)

| Variables | | TC ≥ 200 mg/dL | | | | TG ≥ 150 mg/dL | | | |
|---|---|---|---|---|---|---|---|---|---|
| | | COR (95%CI) | P-value | AOR (95%CI) | p-value | COR (95%CI) | P-value | AOR (95%CI) | P-value |
| Adherence to antihypertensive medicines | Yes | 1 | 0.02 | 1 | 0.03 | 1 | 0.05 | 1 | 0.03 |
| | No | 2.19(1.18–2.89) | | 2.18(1.68–2.83) | | 1.88(1.58–2.33) | | 2.32(1.79–2.92) | |
| Duration of hypertension in year | <5 | 1 | <0.001 | 1 | 0.009 | 1 | 0.01 | 1 | 0.04 |
| | 5–9 | 1.49(1.17–1.97) | | 1.60(0.71–3.60) | | 1.57(1.16–1.90) | | 1.53(0.73–3.21) | |
| | ≥10 | 2.57(1.78–3.13) | | 3.11(1.43–6.77) | | 2.13(1.65–2.98) | | 2.80(1.28–4.95) | |
| Comorbidity with hypertension | No diseases | 1 | <0.001 | 1 | 0.01 | 1 | 0.03 | 1 | 0.02 0.008 0.07 |
| | Renal | 2.15(1.56–4.76) | | 1.59(1.31–2.12) | | 1.75(1.15–2.25) | | 1.60(1.19–8.70) | |
| | Diabetes | 2.83(2.11–6.11) | | 2.86(1.65–3.14) | | 2.56(1.67–9.84) | | 3.29(1.80–4.42) | |
| | Liver | 1.78(1.15–4.13) | | 1.02(0.53–2.11) | | 1.96(1.19–4.83) | | 1.28(0.60–2.75) | |
| Family history of hypercholestronimia | No | 1 | 0.05 | 1 | 0.02 | 1 | 0.04 | 1 | 0.04 |
| | Yes | 1.63(1.11–3.01) | | 1.65(1.13–2.59) | | 1.84(1.21–2.37) | | 2.18(1.29–3.02) | |

Abbreviation: TC: Total cholesterol, TG: Triglycerides, LDL-c: Low-Density Lipoprotein Cholesterol, HDL-c: High-Density Lipoprotein Cholesterol; COR: crude Odds Ratio; AOR: adjusted Odds Ratio, CI: Confidence interval

and 1.8 (AOR: 1.81, 1.41–1.60) times the odds of developing LDL-c and HDL-c than non-smokers, respectively. Those with increased waist circumference were more likely than those with normal waist circumference to have aberrant LDL-c and HDL-c levels (AOR: 1.64, 95% CI: 0.95–2.83) and (AOR: 2.21, 95%CI: 1.28–3.89), respectively. Besides, sedentary lifestyles were associated with 2.5 (AOR: 2.48, 95%CI: 1.18–3.79) and 2.2 (AOR: 2.24, 95%CI: 1.24–4.04) times greater risks of atherogenic LDL-c and HDL-c abnormalities, respectively. Hypertensive patients with DM comorbidity were more likely than those without to have abnormal LDL-c and HDL-c levels by 2.58 (AOR: 2.58, 95%CI: 1.59–4.21) and 2.2 (AOR: 2.20, 95%CI: 1.25–3.74) times, respectively. In addition, those who reported having a family history of hypercholesterolemia had a 1.49 (AOR: 1.49, 95%CI: 1.07–1.91) and 1.90 (AOR: 1.90, 95%CI: 1.31–2.52) times higher risk of LDL-c and HDL-c abnormalities, respectively, compared to those who did not report a family history of hypercholesterolemia (Table 6).

## Discussion

Dyslipidemia was made worse by hypertension's impact on the blood lipid metabolism. Blood lipid concentrations and blood pressure were linked to and impacted by each other, and atherogenic dyslipidemia increased blood pressure variability. In emerging nations, atherogenic dyslipidemia, which is becoming more prevalent, is a significant risk factor for CVD emergence [9, 10]. Majority (93.2%) of the hypertensive patients in this study had at least one lipid profile with atherogenic dyslipidemia, and 17.4% to 38.8% had dyslipidemia with two or more lipid profile derangements. In line with the present study a high prevalence dyslipidemia were reported in Southern part of Ethiopia (90.8%) [28], Lithuania (89.7%) [35], South Africa (85.0%) [36], Poland (77.2%) [37] and Indonesia (79.5%) [38]. However, this finding is higher than previous studies done in Gojjam Ethiopia (48.4%) [27], Mekelle Ethiopia (66.7%) [39], Harar Ethiopia (34.8%) [40], South Africa (67.3%) [41], Uganda (63.3%) [42], Palestine (66.4%) [43] and Iran (30.0%) [44]. In the current study, we recruited only hypertensive patients who were at higher risk for dyslipidemia than general populations. Moreover, this difference might be due to variation in the lifestyles and behavioral characteristics of respondents, sample size, method, stage of urbanization, cut-off values, and socioeconomic status.

Abnormally reduced HDL-c was the most prevalent (59.6%) component of dyslipidemia followed by elevated TG levels (50.3%), which is in line with previous studies [27, 28, 45–48]. Low HDL-c has been linked to atherogenesis and the development of cardiovascular disease, according to data. Most of the patients in our study were at high risk of developing CVD. Besides, the prevalence of elevated LDL-c was 44.3%, which was consistent with earlier studies done in Mekelle, Ethiopia (49.5%) [39], India (47.8%) [49], Iran (50.0%) [50], Thailand (56.5%) [51], Uganda (60.9%) [52], Ghana (61.0%) [53], Senegal (66.3%) [54] and Jordan (74.9%) [55]. The current finding, however, was greater than earlier studies with a similar focus that were conducted in Gojjam (16.1%) [27], and in other regions of Ethiopia (14.1%) [56], and lower than the finding from Southern Ethiopia (60.9%) [28]. The differences in the cutoffs, methodology, respondents' lifestyle, behavioral patterns, and the socioeconomic position may be to blame for these disparities in the results.

The prevalence of elevated total cholesterol (47.7%) is comparable to studies conducted in other African nations [52–54, 56, 57], but greater than those conducted in Harar Ethiopia (33.7%) [40], Southern Ethiopia (38.7%) [28], Mekelle (30.8%) [39], Gojjam (19.6%) [27], and Iran (29.6%) [44]. Interestingly, the prevalence of raised triglycerides (50.3%) in this study was higher than the results from Cameroon (18.9%) [58], Nigeria (9.9%] [59], Ethiopia (21.0%) [56], Malawi (28.7%) [60], Venezuela (39.7%) [61], Jordan (41.9%) [55], and Uganda (42.1%) [52]. This report, however, was lower than the result in Southern Ethiopia (62.2%) [28] but in

**Table 6. Multivariable logistic regression analysis of factors associated with elevated serum LDL-c and reduced HDL-c levels among hypertensive patients in Northeast, Ethiopia, 2021 (n = 384).**

| Variables | | LDL-c > 130 mg/dL | | | | HDL-c <40 mg/dL | | | |
|---|---|---|---|---|---|---|---|---|---|
| | | COR (95%CI) | P-value | AOR (95%CI) | P-value | COR (95%CI) | P-value | AOR (95%CI) | P-value |
| Age in year | 18–40 | 1 | <0.001 | 1 | <0.001 | 1 | 0.02 | 1 | 0.59 |
| | >40 | 6.71(3.95–11.39) | | 4.68(2.0–10.95) | | 1.67(1.09–2.58) | | 1.22(0.58–2.56) | |
| Sex | Male | 1 | 0.02 | 1 | 0.19 | 1 | 0.03 | 1 | 0.014 |
| | Female | 1.76(0.51–2.14) | | 0.71(0.42–1.19) | | 1.63(1.02–1.96) | | 1.55(0.54–2.89) | |
| Residence | Rural | 1 | 0.03 | 1 | 0.01 | 1 | 0.21 | 1 | 0.25 |
| | Urban | 2.80(1.91–3.65) | | 2.01(1.45–3.43) | | 0.77(0.11–1.26) | | 0.77(0.19–1.20) | |
| Marital status | Single | 1 | 0.03 | 1 | 0.84 | 1 | 0.023 | 1 | 0.39 0.35 0.21 |
| | Married | 1.80(1.1–2.96) | | 0.87(0.47–1.61) | | 1.49(0.92–2.41) | | 1.31(0.77–2.22) | |
| | Divorced | 2.58(1.25–5.29) | | 1.06(0.45–2.50) | | 1.24(0.61–2.52) | | 1.12(0.52–2.45) | |
| | Widowed | 2.23(0.91–5.42) | | 0.67(0.24–1.86) | | 2.33(0.90–4.06) | | 2.00(0.71–3.61) | |
| Occupations | Government employee and Student | 1 | 0.002 | 1 | 0.031 | 1 | 0.38 | | |
| | Nongovernment | 1.37(0.72–2.59) | | 1.54(0.81–2.92) | | 0.82(0.44–1.53) | | | |
| | Self employed | 1.96(1.14–3.40) | | 1.45(0.74–2.17) | | 0.84(0.24–2.54) | | | |
| | Farmer | 3.04(1.71–5.39) | | 1.70(0.32–2.85) | | 1.47(0.84–2.92) | | | |
| Monthly income in ETB | >3000 | 1 | 0.38 | | | 1 | 0.89 | | |
| | 2001–3000 | 0.97(0.57–1.65) | | | | 1.24(0.72–2.14) | | | |
| | 1000–2000 | 0.78(0.45–1.38) | | | | 1.08(0.62–1.90) | | | |
| | <1000 | 1.36(0.77–2.41) | | | | 1.10(0.32–1.97) | | | |
| Educational status | Secondary and above | 1 | 0.04 | 1 | 0.07 | 1 | 0.72 | | |
| | Primary | 0.75(0.41–1.36) | | 0.77(0.38–1.58) | | 1.25(0.69–2.26) | | | |
| | Illiterate | 1.33(0.76–2.33) | | 0.90(0.46–1.75) | | 1.24(0.71–2.19) | | | |
| Regular physical activity | Yes | 1 | 0.002 | 1 | 0.02 | 1 | 0.014 | 1 | 0.008 |
| | No | 2.35(1.80–3.28) | | 2.48(1.18–3.79) | | 2.47(1.88–2.45) | | 2.24(1.24–4.04) | |
| Eating habits of fruits/vegetables | Yes | 1 | 0.031 | 1 | 0.04 | 1 | 0.034 | 1 | 0.04 |
| | No | 2.12(1.17–3.78) | | 2.33(1.76–3.30) | | 1.24(0.78–2.01) | | 1.40(.83–2.30) | |
| Body mass index | Underweight | 1 | <0.001 | 1 | 0.001 | 1 | 0.014 | 1 | 0.31 0.03 0.02 |
| | Normal | 1.12(0.33–3.79) | | 0.55(0.13–2.31) | | 1.13(0.27–2.77) | | 1.65(1.19–2.92) | |
| | Overweight | 2.71(0.80–9.23) | | 1.36(0.33–5.71) | | 2.03(1.24–4.48) | | 1.90(1.14–3.66) | |
| | Obesity | 3.40(1.34–5.68) | | 2.1(1.41–7.83) | | 3.13(1.97–5.10) | | 2.71(1.77–4.58) | |
| Waist circumference (cm) (Men/women) | < 94/80 | 1 | 0.008 | 1 | 0.009 | 1 | 0.002 | 1 | 0.004 |
| | ≥94/80 | 1.77(1.16–2.69) | | 1.64(0.95–2.83) | | 2.32(1.86–3.02) | | 2.21(1.28–3.89) | |

(*Continued*)

**Table 6.** (Continued)

| Variables | | LDL-c > 130 mg/dL | | | | HDL-c <40 mg/dL | | | |
|---|---|---|---|---|---|---|---|---|---|
| | | COR (95%CI) | P-value | AOR (95%CI) | P-value | COR (95%CI) | P-value | AOR (95%CI) | P-value |
| Current alcohol consumption | No | 1 | 0.12 | 1 | 0.45 | 1 | 0.79 | | |
| | Yes | 0.65(0.18–1.13) | | 0.92(0.52–1.65) | | 0.93(0.55–1.59) | | | |
| Current cigarette smoking | No | 1 | 0.042 | 1 | 0.05 | 1 | 0.025 | 1 | 0.03 |
| | Yes | 1.96(1.21–2.62) | | 1.81(1.14–3.60) | | 2.00(1.38–2.29) | | 1.75(1.19–2.43) | |
| Habit of chat chewing | No | 1 | 0.03 | 1 | 0.041 | 1 | 0.73 | | |
| | Yes | 2.33(1.77–3.30) | | 1.25(0.65–2.41) | | 0.91(0.53–1.57) | | | |
| Adherence to antihypertensive medicines | Yes | 1 | 0.021 | 1 | 0.042 | 1 | 0.001 | 1 | 0.02 |
| | No | 1.90(1.58–2.36) | | 1.04(0.64–1.70) | | 2.33(1.87–3.05) | | 1.76(1.08–2.20) | |
| Duration of hypertension in year | <5 | 1 | <0.001 | 1 | 0.03 | 1 | 0.04 | 1 | 0.02 0.01 |
| | 5–10 | 4.32(2.42–7.74) | | 1.43(0.63–3.23) | | 1.41(0.90–2.23) | | 1.06(0.50–2.26) | |
| | ≥10 | 4.73(2.89–7.74) | | | | 2.10(1.17–3.74) | | 1.63(0.72–3.66) | |
| Comorbidity with hypertension | No diseases | 1 | 0.02 | 1 | 0.043 | 1 | 0.001 | 1 | 0.04 0.03 0.34 |
| | Renal | 1.21(0.72–2.02) | | 1.96(1.21–2.80) | | 1.03(0.61–1.74) | | 1.02(0.56–1.78) | |
| | Diabetes | 1.58(0.47–5.29) | | 2.58(1.59–4.21) | | 2.81(1.47–4.98) | | 2.20(1.25–3.74) | |
| | Liver | 0.66(0.12–3.64) | | 1.63(1.09–2.24) | | 1.34(0.66–2.89) | | 0.40(0.07–1.32) | |
| Family history of hypercholestronimia | No | 1 | 0.016 | 1 | 0.023 | 1 | 0.001 | 1 | 0.007 |
| | Yes | 1.54(0.33–1.89) | | 1.49(1.07–2.91) | | 2.49(1.69–3.60) | | 1.90(1.13–2.52) | |

Abbreviation: TC: Total cholesterol, TG: Triglycerides, LDL-c: Low-Density Lipoprotein Cholesterol, HDL-c: High-Density Lipoprotein Cholesterol; COR: crude Odds Ratio; AOR: adjusted Odds Ratio, CI: Confidence interval

line with findings reported in Thailand (49.9%) [51], India (56.1%) [49], South Africa (59.3%) [41], and Brazil (65.3%) [62]. Different study populations, methodology, ethnicity, lifestyle, length of hypertension, experiences with antihypertensive medications, and socioeconomic position could all contribute to variations in dyslipidemia prevalence.

Moreover, study discovered that the TC/HDL-C ratio is a strong marker for coronary heart disease. The risk of CVD has been heavily associated with a high TC/HDL-C ratio [63]. Thus, the current investigation found that a higher TC/HDL-c ratio was present in more than half (52.3%) of the study subjects. A study conducted in Karnataka revealed almost similar result (50.0%) [64], while Southern Ethiopia reported a lower result (36.1%) [28]. The difference may be caused by the varying research participant number, the presence of certain illnesses, the amount of dietary consumed, and the utilized cut-off values.

Significant relationships between dyslipidemia and the participant's older age, gender, higher BMI, raised waist circumference, lack of fruit and vegetable consumption, sedentary lifestyle, comorbidity, long-term hypertension for more than five years, non-adherence to anti-hypertensive medications, and current smoking were also found. Dyslipidemia was seen in between 55.0 and 64.0% of people below the age of 40. This is higher than a previous Ethiopian study that revealed 18.7 to 32.5% [27] but lower than research results reported in other publications [65–71]. This increased frequency may be explained by the fact that middle-aged and older people were more vulnerable to the effects of many chronic diseases as physical function declined with age.

Additionally, this study found a positive correlation between elevated waist circumference and dyslipidemia, which is similar with the results of other earlier investigations [72–77]. Derangements in lipid profiles were present in 66.7 to 78.0%. Besides, respondents who were not engaged in regular physical activities experienced serum lipid abnormalities ranging from 52.1% to 66.9%. Similar findings were found in an earlier Ethiopian study that found lipid abnormalities in people who lead sedentary lifestyles to range from 26.4 to 64.4% [28]. Additionally, the prevalence of lipid change among obese, hypertensive people ranged from 76.5% to 91.2%, which is higher than the 32.3% to 56.9% seen in a previous Ethiopian study [27]. Dyslipidemia can develop in people who have sedentary lives, have high BMIs, and have large waist circumferences. Individuals with, sedentary lifestyles, raised BMI, and waist circumference might accumulate excessive fat, which leads to dyslipidemia. Besides, the variation among studies might be due to differences in the study population, methodology, age composition.

The results of the current investigation showed that sex and dyslipidemia were significantly linked. In line with this study, a number of earlier investigations found that the prevalence of dyslipidemia was much greater in women [27, 28, 78, 79]. On the other hand, a prior study found that men had a higher risk of dyslipidemia [80]. Additionally, respondents who had long-term hypertension (66.0 to 76.7%), a family history of hypercholesterolemia (69.2 to 84.6%), and diabetes co-morbidity (60.5 to 72.4%) showed atherogenic dyslipidemia. Dyslipidemia was also strongly linked to non-adherence to antihypertensive medications (55.5 to 73.5%) and infrequent consumption of fruits and vegetables (53.3 to 71.4%). A prior study in Harar, Ethiopia, revealed comparable results regarding consumption of fewer fruits and vegetables [40]. The prevalence of dyslipidemia was substantially correlated with current cigarette smoking, though, with a range of 43.3 to 71.7%, which is consistent with the results of other studies conducted in Saudi Arabia [71].

There were several restrictions placed on the study. The capacity to address causal links between dyslipidemia and its recognized risk variables among hypertension patients is constrained by all cross-sectional study methods, to start with. Second, since the information was gathered by a questionnaire, there could be a bias toward memory.

## Conclusion

It can be concluded from this study that hypertensive patients frequently have atherogenic dyslipidemia, particularly low HDL-c and high triglyceride levels. Besides, the study indicated that age, gender, residence, BMI, the habit of eating fruits/vegetables, current smoking, regular physical activity, duration of hypertension, adhering to antihypertensive medicines, family history of hypercholesterolemia, and comorbidity were predictors of dyslipidemia. Therefore, it is crucial to perform suitable intervention programs aiming at risk factor reduction and establish regular screening programs for blood lipid concentrations in order to combat atherogenic dyslipidemia and the potential development of CVD. We suggest health education programs on behavioral and lifestyle changes for improving the health of hypertension patients based on the findings.

## Supporting information

**S1 Data.**
(SAV)

## Acknowledgments

The authors would like to thank all study participants, data collectors, and all health institutions for their support.

## Author Contributions

**Conceptualization:** Ousman Mohammed.

**Data curation:** Ousman Mohammed, Ermiyas Alemayehu, Mihret Tilahun.

**Formal analysis:** Ousman Mohammed, Endris Ebrahim, Abdurrahman Ali, Hussen Ebrahim, Mihret Tilahun.

**Funding acquisition:** Mesfin Fiseha, Alemu Gedefie, Abdurrahman Ali, Hussen Ebrahim, Mihret Tilahun.

**Investigation:** Ermiyas Alemayehu, Endris Ebrahim, Mesfin Fiseha, Alemu Gedefie, Hussen Ebrahim.

**Methodology:** Ousman Mohammed, Endris Ebrahim, Mesfin Fiseha, Abdurrahman Ali, Hussen Ebrahim, Mihret Tilahun.

**Project administration:** Ousman Mohammed, Ermiyas Alemayehu, Alemu Gedefie, Abdurrahman Ali, Hussen Ebrahim, Mihret Tilahun.

**Resources:** Abdurrahman Ali, Mihret Tilahun.

**Software:** Ousman Mohammed.

**Supervision:** Ermiyas Alemayehu, Endris Ebrahim, Mesfin Fiseha, Alemu Gedefie, Abdurrahman Ali, Hussen Ebrahim.

**Validation:** Ousman Mohammed, Alemu Gedefie.

**Visualization:** Ousman Mohammed, Ermiyas Alemayehu, Endris Ebrahim, Mesfin Fiseha, Alemu Gedefie, Abdurrahman Ali, Hussen Ebrahim, Mihret Tilahun.

**Writing – original draft:** Ousman Mohammed, Ermiyas Alemayehu, Endris Ebrahim, Mesfin Fiseha, Alemu Gedefie, Hussen Ebrahim, Mihret Tilahun.

**Writing – review & editing:** Ousman Mohammed, Abdurrahman Ali, Mihret Tilahun.

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
