## [Decision Letter · Decision Letter 0]

9 Aug 2022

PONE-D-22-18116Atherogenic dyslipidemia and associated risk factors among hypertensive patients of five health facilities in Northeast EthiopiaPLOS ONE

Dear Dr. Ousman Mohammed,

Thank you for submitting your manuscript to PLOS ONE. After careful consideration, we feel that it has merit but does not fully meet PLOS ONE’s publication criteria as it currently stands. Therefore, we invite you to submit a revised version of the manuscript that addresses the points raised during the review process. Please especially pay attention to comments by Reviewer 2. They need to be fully addressed. Please submit your revised manuscript by Sep 23 2022 11:59PM. If you will need more time than this to complete your revisions, please reply to this message or contact the journal office at plosone@plos.org. Please include the following items when submitting your revised manuscript:A rebuttal letter that responds to each point raised by the academic editor and reviewer(s). You should upload this letter as a separate file labeled 'Response to Reviewers'.A marked-up copy of your manuscript that highlights changes made to the original version. You should upload this as a separate file labeled 'Revised Manuscript with Track Changes'.An unmarked version of your revised paper without tracked changes. You should upload this as a separate file labeled 'Manuscript'.If applicable, we recommend that you deposit your laboratory protocols in protocols.io to enhance the reproducibility of your results. Protocols.io assigns your protocol its own identifier (DOI) so that it can be cited independently in the future. For instructions see: https://journals.plos.org/plosone/s/submission-guidelines#loc-laboratory-protocols. Additionally, PLOS ONE offers an option for publishing peer-reviewed Lab Protocol articles, which describe protocols hosted on protocols.io. Read more information on sharing protocols at https://plos.org/protocols?utm_medium=editorial-email&utm_source=authorletters&utm_campaign=protocols.

We look forward to receiving your revised manuscript.

Kind regards,

Paolo Magni

Academic Editor

PLOS ONE

Journal Requirements:

"The authors declare that they have no potential competing interests."

Additional Editor Comments:

The paper requires some polishing of the language and some details still need to be improved according to Reviewer 2.

Reviewers' comments:

Reviewer's Responses to Questions

**Comments to the Author**

1. Is the manuscript technically sound, and do the data support the conclusions?

Reviewer #1: Yes

Reviewer #2: Yes

2. Has the statistical analysis been performed appropriately and rigorously? 

Reviewer #1: Yes

Reviewer #2: Yes

3. Have the authors made all data underlying the findings in their manuscript fully available?

Reviewer #1: Yes

Reviewer #2: No

4. Is the manuscript presented in an intelligible fashion and written in standard English?

Reviewer #1: Yes

Reviewer #2: Yes

5. Review Comments to the Author

Reviewer #1: A well written report of an interesting topic in Ethiopia. The manuscript is technically sound and all the data support the conclusions. The statistical analysis has been performed appropriately and rigorously. Well done.

Reviewer #2: This manuscript answers a relevant research question and uses the correct methodology to do so.

While it should be clear that associations found through the logistic regression are exploratory in nature, with the study not powered to consider this amount of predictors, the level of proof is generally high.

The manuscript is of decent quality and almost ready for publication as far as I am concerned, although some improvements and clean-up remains to be done.

Please find my suggestions for updates below.

Major comments

(1) The primary objective and most important outcome of this study is the percentage of hypertensive people with dyslipidemia. The sample size was calculated to estimate this with a specific precision.

Yet, no confidence intervals are presented around the point estimates of the actual percentage of hypertensive people with dyslipidemia. Such CIs are useful as they indeed show how precise the estimates for prevalence of the different dyslipidemia markers are.

(2) Wording referring to the population (results, discussion):

Sometimes “participants” is used, other times “people” or identifiers like “women”.

None of these are technically correct. The population here is “people with hypertension in Northeast Ethiopia”, which is different from the sample (a subset of such people) or from “people” in general.

While it doesn’t need to be repeated constantly, some attention should be given throughout the manuscript to the fact that this analysis is representative for the proper population: people with hypertension in Northeast Ethiopia.

This is correctly done in the abstract and conclusion.

(3) Material & Methods: sometimes too much details are given and information tends to get needlessly repeated

Examples: definition of BMI (repeated twice, and common knowledge so not needed), multiple repeats on how height measurements are taken, paragraph on quality control can probably be shortened, …

(4) Table 5: please format the table better, it is hard to read like this.

Table 5 and Table 6: please doublecheck all numbers carefully. Some results, while possible, are a bit strange.

It is unclear to me how confidence intervals for logistic regression are calculated in SPSS, these results do not match what other statistical software could give.

A few examples: 2.02(1.62-3.70) (maybe the lower bound is 1.26?); 1.33(1.08-1.40) , 2.86(1.65-3.14) (does not make sense that the estimate is closer to the upper bound in these cases).

Since it’s very easy to introduce typos when copying the results from the statistical package to the manuscript, authors should carefully doublecheck every number.

(5) Conclusion: “were significant determinants of dyslipidemia”

Alas, you did it perfect during the whole paper, even correctly explicitly mentioning that a cross-sectional study cannot address causal links.

Since no causality can be concluded, you should not state that these are “determinants” of dyslipidemia. Instead using “predictors” or “significantly associated” are both correct options.

(6) I did not find a data sharing statement in the manuscript. Instead, when talking about data sharing, you replied "No - some restrictions will apply" and data available "On request". I understand that this is not a sufficiently detailed answer in light of the PLOS Data Policy. Please provide clear instructions on how a sufficiently anonymised version of the data can be obtained.

Minor comments:

(7) Abstract and page 7 (Data analysis):

“Statistical significance was declared at p ≤0.05.”

Please change to p < 0.05 (shouldn’t make a difference in results or interpretation).

(8) Page 5: (Study Population and legibility Criteria)

“All volunteer hypertensive patients (…) were eligible. However (…)”

Please rewrite. The part following “however” makes it clear that not all patients were eligible.

(9) Page 5: (Data collection and quality control)

“Well-trained two data collectors”

Rewrite to “Two well-trained data collectors”

(10) Table 1:

HDL-c <40 mg/dL vs. occupation: p-value is missing for this specific comparison

(11) Page 11, bottom:

“the habit of not eating fruits and vegetables were substantially linked with elevated TC levels but not with elevated TG levels (Table 5).”

Replace substantially by significantly. Substantial would be linked to effect size, while this statement should only refer only to statistical significance.

(12) Page 14:

“(AOR: 1.49, 95%CI: 127-1.91)”: add decimal sign in 1.27

“and 2.0 (AOR: 1.90, 95%CI: 1.51-2.52)”: change 2.0 by 1.9

6. PLOS authors have the option to publish the peer review history of their article (what does this mean?). If published, this will include your full peer review and any attached files.

Reviewer #1: No

Reviewer #2: No

---

## [Author Response · Author response to Decision Letter 0]

16 Aug 2022

Dear editors and reviewers, I greatly acknowledge your comments which are actually very crucial for the quality of the paper as well as for my future improvements. As per your comments, all things are corrected and the corrected parts were alighted with green.

Major comments

Reviewer point #1): The primary objective and most important outcome of this study is the percentage of hypertensive people with dyslipidemia. The sample size was calculated to estimate this with a specific precision. Yet, no confidence intervals are presented around the point estimates of the actual percentage of hypertensive people with dyslipidemia. Such CIs are useful as they indeed show how precise the estimates for prevalence of the different dyslipidemia markers are.

Author response: Yes, we agreed with the reviewers, a 95% CI was calculated and indicated in the result part. Here is the calculated CI (93.2%; 95%CI: 90.6-95.6)

Reviewer point #2): Wording referring to the population (results, discussion): Sometimes “participants” is used, other times “people” or identifiers like “women”. None of these are technically correct. The population here is “people with hypertension in Northeast Ethiopia”, which is different from the sample (a subset of such people) or from “people”ingeneral. While it doesn’t need to be repeated constantly, some attention should be given throughout the manuscript to the fact that this analysis is representative for the proper population: people with hypertension in Northeast Ethiopia. This is correctly done in the abstract and conclusion.

Author response: Of course, this study was inferred to the hypertensive individuals so, as per the reviewer's comment we tried to replace some identifiers like “women, people, and participants with more appropriate descriptions like hypertensive individuals or people with hypertension.

Reviewer point #3): Material & Methods: sometimes too much details are given and information tends to get needlessly repeated Examples: definition of BMI (repeated twice, and common knowledge so not needed), multiple repeats on how height measurements are taken, paragraph on quality control can probably be shortened,

Author response: Here we accepted the reviewer's comment and it was due to clerical errors. Now we avoid repetitive descriptions and unnecessary details and it was shown in the manuscript. 

Reviewer point #4): Table 5: please format the table better, it is hard to read like this. Table 5 and Table 6: please doublecheck all numbers carefully. Some results, while possible, are a bit strange. It is unclear to me how confidence intervals for logistic regression are calculated in SPSS, these results do not match what other statistical software could give. A few examples: 2.02(1.62-3.70) (maybe the lower bound is 1.26?); 1.33(1.08-1.40), 2.86(1.65-3.14) (does not make sense that the estimate is closer to the upper bound in these cases). 

 Author response: Based on the reviewer’s suggestion we tried to check all the numbers carefully. Since it was an error while copying numbers and now it was solved and corrected. 

 Reviewer point #5): Conclusion: “were significant determinants of dyslipidemia” Alas, you did it perfect during the whole paper, even correctly explicitly mentioning that a cross-sectional study cannot address causal links. Since no causality can be concluded, you should not state that these are “determinants” of dyslipidemia. Instead using “predictors” or “significantly associated” are both correct options.

Author response: Yes we accept the reviewer's comment and we corrected it following the reviewer's suggestion.

Reviewer point #6): I did not find a data sharing statement in the manuscript. Instead, when talking about data sharing, you replied "No - some restrictions will apply" and data available "On request". I understand that this is not a sufficiently detailed answer in light of the PLOS Data Policy. Please provide clear instructions on how a sufficiently anonymised version of the data can be obtained.

Author response: It was an error while submitting the manuscript online and I can assure you that all relevant data are included in the paper. 

Minor comments:

Reviewer point #7): Abstract and page 7 (Data analysis): “Statistical significance was declared at p ≤0.05.” Please change to p < 0.05 (shouldn’t make a difference in results or interpretation).

Author response: Yes we have corrected p ≤0.05 was changed to p < 0.05 

Reviewer point #8): Page 5: (Study Population and legibility Criteria) “All volunteer hypertensive patients (…) were eligible. However (…)” Please rewrite. The part following “however” makes it clear that not all patients were eligible.

Author response: Some physiological and pathological factors that could alter serum lipid profiles, such as pregnancy, taking lipid-altering drugs, and antihyperlipidemic medications, were excluded because the study was designed to assess dyslipidemia. 

Reviewer point #9): Page 5: (Data collection and quality control) “Well-trained two data collectors” Rewrite to “Two well-trained data collectors”

Author response: Yes we have to rewrite it as per the reviewer's comment.

Reviewer point #10): Table 1: HDL-c <40 mg/dL vs. occupation: p-value is missing for this specific comparison

Author response: The missing p-value was putt as 0.21 in the space provided.

Reviewer point #11): Page 11, bottom: “the habit of not eating fruits and vegetables were substantially linked with elevated TC levels but not with elevated TG levels (Table 5).”

Replace substantially by significantly. Substantial would be linked to effect size, while this statement should only refer only to statistical significance.

Author response: Here we have replaced the word ‘substantially’ with significantly and you can see it in the manuscript. 

Reviewer point #12): Page 14: “(AOR: 1.49, 95%CI: 127-1.91)”: add decimal sign in 1.27 “and 2.0 (AOR: 1.90, 95%CI: 1.51-2.52)”: change 2.0 by 1.9

Author response: Yes it was corrected.

---

## [Decision Letter · Decision Letter 1]

21 Sep 2022

PONE-D-22-18116R1Atherogenic dyslipidemia and associated risk factors among hypertensive patients of five health facilities in Northeast EthiopiaPLOS ONE

Dear Dr. Ousman Mohammed,

Thank you for submitting your manuscript to PLOS ONE. After careful consideration, we feel that it has merit but does not fully meet PLOS ONE’s publication criteria as it currently stands. Therefore, we invite you to submit a revised version of the manuscript that addresses the points raised during the review process.

Please address the reviewers'comments.

We look forward to receiving your revised manuscript.

Kind regards,

Paolo Magni

Academic Editor

PLOS ONE

Journal Requirements:

Additional Editor Comments (if provided):

Some aspects need some improvement.

Reviewers' comments:

Reviewer's Responses to Questions

**Comments to the Author**

1. If the authors have adequately addressed your comments raised in a previous round of review and you feel that this manuscript is now acceptable for publication, you may indicate that here to bypass the “Comments to the Author” section, enter your conflict of interest statement in the “Confidential to Editor” section, and submit your "Accept" recommendation.

Reviewer #1: All comments have been addressed

Reviewer #2: (No Response)

2. Is the manuscript technically sound, and do the data support the conclusions?

Reviewer #1: Yes

Reviewer #2: Yes

3. Has the statistical analysis been performed appropriately and rigorously? 

Reviewer #1: Yes

Reviewer #2: Yes

4. Have the authors made all data underlying the findings in their manuscript fully available?

Reviewer #1: Yes

Reviewer #2: No

5. Is the manuscript presented in an intelligible fashion and written in standard English?

Reviewer #1: Yes

Reviewer #2: Yes

6. Review Comments to the Author

Reviewer #1: Well done on your study.

The introduction is well written, the methods are adequately described and the discussion/conclusion is supported by the results.

Maybe it could be worthy to amend the design of the tables to increase readability.

Reviewer #2: Please review the data availability policy. https://journals.plos.org/plosone/s/data-availability

It is still unclear to me whether the data source, or an anonymised version thereof will be made freely available. The statement "Yes - all data are fully available without restriction" and the reply to the comment that "all relevant data are included in the paper" are contradictory.

"All data" would at minimum be a dataset with a single line per patient that was included in the study and a single column for each variable that was reported in this manuscript. No instructions on how or where to obtain such dataset are provided.

7. PLOS authors have the option to publish the peer review history of their article (what does this mean?). If published, this will include your full peer review and any attached files.

Reviewer #1: No

Reviewer #2: No

---

## [Author Response · Author response to Decision Letter 1]

22 Sep 2022

Dear Editors and Reviewers, 

As per your comments, everything was revised and fixed in the last submission. However, aside from the Data Availability Statement concern expressed by Reviewer 2, I couldn't find any particular concerns that still needed to be fixed and submitted with tracked modifications. I stated that there are some restrictions in my initial proposal. Naturally, it was a mistake that was done by the correspondence author alone, who neglected to pay attention during the online submission process. However, I had highlighted it in my prior responses to reviewers when I stated, "All the essential data are within the manuscript." Therefore, I have also included the Data Availability Statement by saying ‘All the relevant data are within the manuscript’ in the manuscript. 

Thank you very much for your consideration. 

Your Sincerely,

---

## [Decision Letter · Decision Letter 2]

17 Oct 2022

PONE-D-22-18116R2Atherogenic dyslipidemia and associated risk factors among hypertensive patients of five health facilities in Northeast EthiopiaPLOS ONE

Dear Dr. Mohammed,

Thank you for submitting your manuscript to PLOS ONE. After careful consideration, we feel that it has merit but does not fully meet PLOS ONE’s publication criteria as it currently stands. Therefore, we invite you to submit a revised version of the manuscript that addresses the points raised during the review process.

Specifically, please address the comment by the reviewer: "The data statement refers to all data being contained in the manuscript, this is not correct.".

We look forward to receiving your revised manuscript.

Kind regards,

Paolo Magni

Academic Editor

PLOS ONE

Journal Requirements:

Additional Editor Comments:

The Authors just need to address the comment by the reviewer (The data statement refers to all data being contained in the manuscript, this is not correct.).

Reviewers' comments:

Reviewer's Responses to Questions

**Comments to the Author**

1. If the authors have adequately addressed your comments raised in a previous round of review and you feel that this manuscript is now acceptable for publication, you may indicate that here to bypass the “Comments to the Author” section, enter your conflict of interest statement in the “Confidential to Editor” section, and submit your "Accept" recommendation.

Reviewer #2: (No Response)

2. Is the manuscript technically sound, and do the data support the conclusions?

Reviewer #2: Yes

3. Has the statistical analysis been performed appropriately and rigorously? 

Reviewer #2: Yes

4. Have the authors made all data underlying the findings in their manuscript fully available?

Reviewer #2: No

5. Is the manuscript presented in an intelligible fashion and written in standard English?

Reviewer #2: Yes

6. Review Comments to the Author

Reviewer #2: Please refer to my previous review. The data statement refers to all data being contained in the manuscript, this is not correct. Please discuss with the editor how data will be made available and/or which restrictions will apply.

7. PLOS authors have the option to publish the peer review history of their article (what does this mean?). If published, this will include your full peer review and any attached files.

Reviewer #2: No

---

## [Author Response · Author response to Decision Letter 2]

17 Oct 2022

Dear Editors and Reviewers, 

Reviewer point: Have the authors made all data underlying the findings in their manuscript fully available?

Author response: Pardon me for the inconvenience. As per your comments, everything was revised and fixed in the last submission. However, I stated that there are some restrictions in my initial submission; naturally, it was a mistake that was done by the correspondence author alone, who neglected to pay attention during the online submission process. Now I assure you again there is no restriction regarding data availability at all. Therefore, I have also included the Data Availability Statement by saying; we will provide repository information for our data at acceptance as Supporting Information files. And I highlighted it with yellowish. 

Editor point: Journal Requirements:

Author response: I have already checked all the references, and it is complete and correct. 

Thank you very much for your consideration. 

Kind regards, 

Ousman

---

## [Decision Letter · Decision Letter 3]

24 Oct 2022

Atherogenic dyslipidemia and associated risk factors among hypertensive patients of five health facilities in Northeast Ethiopia

PONE-D-22-18116R3

Dear Dr. Mohammed,

We’re pleased to inform you that your manuscript has been judged scientifically suitable for publication and will be formally accepted for publication once it meets all outstanding technical requirements.

Kind regards,

Paolo Magni

Academic Editor

PLOS ONE

Additional Editor Comments (optional):

All comments have been addressed.

Reviewers' comments:

Reviewer's Responses to Questions

**Comments to the Author**

1. If the authors have adequately addressed your comments raised in a previous round of review and you feel that this manuscript is now acceptable for publication, you may indicate that here to bypass the “Comments to the Author” section, enter your conflict of interest statement in the “Confidential to Editor” section, and submit your "Accept" recommendation.

Reviewer #2: All comments have been addressed

2. Is the manuscript technically sound, and do the data support the conclusions?

Reviewer #2: Yes

3. Has the statistical analysis been performed appropriately and rigorously? 

Reviewer #2: Yes

4. Have the authors made all data underlying the findings in their manuscript fully available?

Reviewer #2: Yes

5. Is the manuscript presented in an intelligible fashion and written in standard English?

Reviewer #2: Yes

6. Review Comments to the Author

Reviewer #2: (No Response)

7. PLOS authors have the option to publish the peer review history of their article (what does this mean?). If published, this will include your full peer review and any attached files.

Reviewer #2: No

---

## [Editor Report · Acceptance letter]

11 Nov 2022

PONE-D-22-18116R3 

Atherogenic dyslipidemia and associated risk factors among hypertensive patients of five health facilities in Northeast Ethiopia  

Dear Dr. Mohammed:

I'm pleased to inform you that your manuscript has been deemed suitable for publication in PLOS ONE. Congratulations! Your manuscript is now with our production department. 

Kind regards, 

on behalf of

Prof. Paolo Magni 

Academic Editor

PLOS ONE